# Prevalence of Pediatric and Adolescent Balance Disorders: Analysis of a Mono-Institutional Series of 472 Patients

**DOI:** 10.3390/children8111056

**Published:** 2021-11-16

**Authors:** Cristiano Balzanelli, Daniele Spataro, Luca Oscar Redaelli de Zinis

**Affiliations:** 1Vertigo Center—San Bernardino Polyclinic of Salò, 25087 Salò, Italy; balzanelli.critiano@gmail.com; 2ENT Department—ARNAS Garibaldi of Catania, 95100 Catania, Italy; spataro.daniele.sd@gmail.com; 3Department of Medical and Surgical Specialties, Radiological Sciences, and Public Health, Section of Audiology, University of Brescia, 25100 Brescia, Italy; 4Pediatric Otolaryngology Head Neck Surgery Department, Children Hospital, ASST Spedali Civili, 25100 Brescia, Italy

**Keywords:** child, adolescent, vertigo, dizziness, differential diagnosis

## Abstract

(1) Background: To assess the prevalence and frequency distribution of balance disorders in children and adolescents to delineate the planning of a targeted clinical and instrumental diagnostic work-up; (2) Methods: Retrospective analysis of the clinical documentation of patients under 18 years suffering from balance disorders from 2010 to 2019. Detailed collection of clinical history, accurate clinical examination, including both nystagmus and vestibulospinal signs examinations, and specific instrumental testing were the basis of the diagnostic process. (3) Results: A total of 472 participants were included in the study. Vestibular loss (26.1%) was the most frequent cause of vertigo in children, followed by vestibular migraine (21.2%) and benign paroxysmal positional vertigo (10.2%). In 1.1% of patients, the cause of vertigo remained undefined; (4) Conclusions: The diagnostic process applied was effective in understanding the cause of balance disorders in most cases and prevents more complex and expensive investigations reserved for only a few selected cases.

## 1. Introduction

Vertigo in a pediatric age is not an infrequent condition, with about a 10% prevalence in most clinical trials [1,2,3,4,5,6,7,8,9,10,11,12,13]. The causes, as for adults, can be highly variable, especially when related to functional, visual, vestibular, or proprioceptive somesthetic system dysfunction, or to syndromic/not-syndromic genetic/hereditary type diseases associated with atypical neural development [14]. Central nervous system or vestibular defects, at any age of a child, can cause somatosensory, cognitive, and motor output disorganization, incorrect postural control, lack of coordination, or imbalance of variable intensity and duration [15,16].

The diagnostic assessment of vertiginous syndrome in pediatric patients is difficult because of their reduced communication skills, the capacity to be easily distracted and the inability to distinguish vertigo from fear [4]. The variability of the clinical presentation of vertigo justifies a multi-specialist diagnostic approach, involving not only a neurotologist but, in selected cases, also a psychiatrist, psychologist, cardiologist, pediatrician, geneticist, traumatologist, physiatrist, ophthalmologist, and a dentist [17].

The prevalence data of the different forms of vertigo studied in the scientific literature are highly variable due to the heterogeneity of the examined samples, the different methods of clinical evaluation or statistical processing of the data, and the different cultural settings of the authors (neurologist, pediatrician, otolaryngologist) [14].

This study aims to assess the prevalence and frequency distribution of balance disorders in our institution to delineate the planning of a targeted clinical and instrumental diagnostic work-up, thus ensuring adequate management of vertigo in children.

## 2. Materials and Methods

This paper was observational, retrospective, and non-profit. The cohort of patients was obtained by performing an analysis of the available clinical documentation (computer and paper medical records) to identify patients under 18 years old, suffering from balance disorders, who were tested at our Institution from 2010 to 2019.

Exclusion criteria were patients with symptoms not related to vertigo, subjects examined for other reasons (children sent for hearing impairment, i.e., before cochlear implant surgery), and patients not accompanied by parents, relatives, or legal guardians.

The database shows all clinical data regarding age at the time of the first examination, type and number of vertigo episodes, associated symptoms (nausea/vomiting, hearing loss, headache), familial migraine vertigo, history of motion sickness, and general health problems, before other specialistic or diagnostic tests, until the end of 2020.

All patients were subjected to complete clinical examination of the vestibular function by infrared video-nystagmoscopy (Synapsis Nystalab—Frenzel wireless video system), including:Study of eye movements (horizontal and vertical convergence, saccades, smooth pursuit, fixation);Spontaneous nystagmusPosition nystagmus (NyPos) (supine/Rose position, right, and left side position);Positioning nystagmus (NyPP) (Dix-Hallpike maneuver and Roll-supine test);Evocative nystagmus (Head Shaking Test and Head Impulse Test).

Vestibulo-spinal reflexes were assessed by the Romberg test, Unterberger/Fukuda test, and stretched arms test. Because younger children (3–4 years old) can be easily distracted and may not always cooperate, we analyzed spontaneous head-neck-superior-inferior arms movements, postural attitude, strength and coordination during spontaneous playing, walking, running, and interaction with parents.

If clinical vestibular evaluation was negative, rotatory chair test (RCT) was performed until 2012, whereas from 2013 patients were also submitted to video head impulse test (VHIT) (Natus-Otometrics, Godtrup, Denmark) to assess amplitude of the vestibulo-ocular reflex (VOR) including conventional head impulse test paradigm (HIMP), suppression head impulse paradigm (SHIMP), and the presence of compensatory/anti-compensatory saccades at high frequency stimuli in all semicircular canals. No patient underwent vestibular caloric stimulation (VCS), due to its invasiveness and poor tolerability. No patient underwent vestibular evoked myogenic potentials (VEMPs), due to the high degree of movement artifacts and low electrode impedance, which are often present in pediatric patients.

All children were subjected to otoscopy, audiological evaluation with tympanometry, and in most cases a pure tone audiometry was carried out.

When vestibular clinical and/or instrumental testing were inconclusive, the children were submitted to an accurate interview with the parents and to a child neuropsychiatric evaluation, to assess their degree of neurodevelopment and to exclude sensory-motor deficiencies. Based on the neurologic outcome or in case of a still inconclusive report, other diagnostic investigations were performed such as auditory brainstem response (ABR), electro-encephalogram (EEG), computed tomography (CT) scan and magnetic resonance imaging (MRI) of the brain, electrocardiogram (ECG) or neuropsychiatric, ophthalmological, cardiological, and dental evaluation.

ICHD-3 (International Classification of Headache Disorders) diagnostic criteria of 2018 [18] were considered for benign paroxysmal vertigo of children (BPVC) and vestibular migraine diagnosis. Guidelines of the AAO-HNS—American Academy of Otolaryngology-Head and Neck Surgery Committee of 1995, revisited in 2015 [19,20], were considered for diagnosis of Ménière’s disease. Psychogenic vertigo was classified considering ICD-10 Criteria, after Neuropsychiatric consultation [21].

Follow-up varied from 1 to 10 years.

The study was approved by the ethics committee of our institution (protocol number 83787). Informed written consent was obtained from the parents of each participant.

## 3. Results

A total of 472 participants were considered in the study. There were 281 (60%) females and 191 (40%) males with a mean age of 11 ± 3 years at the time of the first observation. In our sample, vestibular loss (26.1%) was the most frequent cause of vertigo in children, followed by vestibular migraine (21.2%) and benign paroxysmal positional vertigo (BPPV) (10.2%). BPVC was present in 8.5% cases (40/472), especially in females (67.5% -27/40- vs. 32.5% -13/40-) and in children under the age of 7 (52.7% -29/55- vs. 2.6% -11/417-) with a mean age of 5 ± 3 years. Psychogenic vertigo was found in 8.5% of our sample. We found ophthalmological disorders in 4.9% of cases, dental disorders in 4.4%, hemodynamic orthostatic disorders in 4.0%, otitis media in 3.2%, epileptic disorders in 2.5%, postural disorders in 2.3%, trauma in 1.5%, Meniere’s Disease in 0.8%, other neurological diseases (L-Dopa responsive dystonia, outcome of brain hemorrhage, and multiple sclerosis) in 0.6%. In 1.1% of patients, the cause of vertigo remained undefined. The distribution of causes in the entire group and according to gender is reported in Table 1.

A family history for migraine was present in 57% of vestibular migraine patients and in 35% of BPVC patients. Previous headache episodes were reported in 77% of vestibular migraine patients, motion sickness was found in 30% of vestibular migraine patients, and neurovegetative symptoms (nausea, vomiting, cold sweats, and pallor) were reported in 42% cases of vestibular loss. Hearing disorders (fullness, tinnitus, and hearing loss) were found in 75% of Meniere’ cases. Ophthalmological disorders were observed in 9% of vestibular migraine patients. Family history of migraine and complete distribution of symptoms according to the cause of vertigo in our sample is reported in Table 2.

Spontaneous nystagmus was observed in 20.8% cases of vestibular loss (26/123) and 50% of Meniere’s disease (2/4). Positional nystagmus was found in only 13.6% cases of vestibular loss (17/123). Positioning paroxysmal nystagmus was recorded in 47.9% cases of BPPV. Vestibulospinal signs were present in 75% of Meniere’s disease, 66.6% of neurologic disorders, 29.6% of vestibular loss, 27.3% of postural disorders, and 25% of epileptic disorders. Clinical examination results according to the cause of vertigo are reported in Table 3.

Abnormal results of RCT were observed in 89.7% of those with vestibular loss and 100% with Meniere’s disease. Abnormal results of VHIT, including reduced HIMP and SHIMP VOR amplitude and/or presence of refixation overt/covert saccades, were observed in 71.6% of vestibular loss. In all other conditions, RCT or VHIT did not add further diagnostic information. The results of instrumental examinations according to the cause of vertigo are reported in Table 4.

## 4. Discussion

Vertigo is not an infrequent condition in the pediatric age and, although there is wide variability in published prevalence data, approaches nearly 10% in most clinical trials [1,2,3,4,5,6,7,8,9,10,11,12,13]. As in adults, control of standing position and harmonic movements in children depend on the integration of many sensory activities (vestibular, visual, proprioceptive, hemodynamic, and cognitive). Anatomic, functional, or psychological abnormalities in different stages of children’s neurodevelopment, or any alteration of their sensory-motor integration, justify the variability of different clinical and instrumental published reports [22,23]. Small coordination issues or slight delays in motor development are at the basis of balance and walking disorders [24]. Late, missed, or inadequate neurotologic evaluation causes impairment of children’s daily activities, since they tend to isolate themselves and avoid normal recreational activities, with emotional, affective, physiological, and behavioral impairment, which can be permanent and occasionally influence future relationships [25].

In the pediatric age, vertigo is better tolerated and characterized by soft and short-lived symptoms, thanks to their low negative cognitive experience and highly effective neuronal plasticity [24]. Sometimes they can even enjoy themselves during vestibular stimulations, as it is perceived like rides or playing video games (eyes stimulation, head accelerations, quick turn and stop of the body); however, in case of illness children get frightened and feel fear and discouragement, since they are not able to get control of the situation [26]. When they experience vertigo, they feel panicked, generally causing both families and physicians to worry. Lack of knowledge in children about the causes and prevalence, balance disorders, correlation with age and gender, standard diagnostic work-up, often lead to inappropriate, invasive, or expensive exams (i.e., CT or RMN scan), with no useful contribution towards proper therapeutic management [14].

As reported in the published literature, the actual incidence of vertigo in pediatric patients remains underestimated, due to the lack of correct clinical classification and diagnostic work-up [14]. Moreover, symptoms can go unnoticed in very young children, as they are unable to communicate their discomfort, but other symptoms can be easily detected, such as vomiting, pallor, abdominal pain, and ataxia [4].

In most clinical reports, differences in terms of prevalence of the single pathologies that cause vertigo can be observed [24]. It is estimated that 5–10% of children have at least one episode of vertigo before the age of 10 years, with 10% having at least one episode per year; in 51.5% of cases the intensity of vertigo stops ongoing activities and in 25 to 50% of cases vertigo is associated with a form of migraine [2,5,6,8,11]. In our sample, familial migraine was observed in 24.2% (114/472) of patients, headache episodes in 35.0% (165/472), and motion sickness in 10.0% (47/472).

We also compared the data in our study with that in a recent review which examined a very large sample of 2726 patients aged between 2 months and 19 years [27]. In this review, the main four diagnoses associated to vertigo accounted for 57% of cases, including vestibular migraine (23.8%), BPVC (13.7%), unidentified cause (11.7%), and vestibular labyrinthitis/neuritis (8.47%). Less common diagnoses included Ménière’s disease (3.0%) and tumors of the central nervous system (1.2%) [27].

In our experience, the four most frequent diagnoses of vertigo in patients aged 2 months—18 years accounted for 66% of cases, including respectively unilateral vestibular loss (26.1%), vestibular migraine (21.2%), BPPV (10.2%), and BPVC (8.5%). Psychogenic vertigo was found in 8.5% of cases. Furthermore, we found other causes in 24.2% of cases (epilepsy, neurological diseases, bilateral vestibulopathy, otitis media with effusion, ophthalmologic and dental disorders, orthostatic hypotension, trauma, Ménière’s disease, and postural disorders). The cause could not be demonstrated in 1.1% of cases (Table 1).

According to the published literature, our data confirm that vertigo and balance disorders in the pediatric age is a benign condition in most cases [14]. We found that other neurological diseases are present in only 0.6% of patients (3/472): L-Dopa responsive dystonia, outcome of brain hemorrhage, and multiple sclerosis.

Moreover, we noticed a significant discrepancy between the published percentage of vestibular labyrinthitis/neuritis (8.47%) [27] and unilateral vestibular loss in our sample of patients (26.1%), in both males and females. This is probably due to our combined and almost systematic vestibular clinical plus instrumental diagnostic work-up (infrared videonystagmoscopy and/or video HIT and/or RCT). Consequently, we obtained a consistent reduction in the percentage of vertigo for unknown causes in only 1.1% of cases, compared to published data where it is more than 10% [27].

As previously indicated, in our sample the most frequent cause of vertigo was vestibular neuritis (26.1%) (Table 1), in discordance with general published data, where the most common clinical diagnosis of vertigo in children at any age is vestibular migraine [24,28]. In this frequent condition, it is described that patients can present both central and peripheral vestibular signs in 73% of cases [29]. Intense motion sickness generally appears at age 2–3 and familial migraine is described in over 70% of cases [30]. Our sample confirms that 57% of children with vestibular migraine had a history of familial migraine, where 77% suffered from previous headache and only 19% had a history of motion sickness. Davitt’s data show that 44.4% of vestibular migraine patients had an ophthalmologic disorder that simultaneously worsened symptoms and control of migraine events (i.e., vergence dysfunction or refraction defects) [31]. On the contrary, only 9% of children in our sample showed simultaneous vestibular migraine and ophthalmological disorders. Moreover, it is well known that children with vertigo and normal vestibular function can present various vergence dysfunctions causing dizziness from 5–6 years of age [32,33,34]. In our sample, 4.9% of cases with a mean age of 11 ± 3 years was due to visual disturbances (Table 1). These disorders usually appear after 6 years of age, when activities requiring prolonged attention are more frequent and their incidence is constantly increasing due to the greater use of devices equipped with video screens, which determine an excessive effort on ocular motility [29,35]. These never cause serious rotational vertigo, but feelings of brief and repeated rotations or pitching are often related to visual fatigue. Symptoms can also appear when awakening, in particular for hypermetropia conditions. It has been shown that these children are more prone to postural instability, with resolution of symptoms by ophthalmological treatment in two-thirds of cases [33,34]. The evidence therefore suggests that complete eye examination should be considered for children affected by any imbalance or vertigo symptoms.

The prevalence of BPVC in the published literature varies from 6 to 20% [31,32]. In our sample, BPVC also represented the most common cause of vertigo under the age of 7 (52.7% vs. 2.6%) with a mean age of 5 ± 3 years. In all cases we observed spontaneous recovery of BPVC within the range of 7 to 10 years of age. Seventy percent of patients with BPVC develop a typical migraine in adolescence, which meets the ICHD III criteria. This is part of the “*Childhood Periodic Syndromes*” together with the paroxysmal torticollis and recurring gastrointestinal disorder (cyclic vomiting and abdominal migraine), and represents an early migraine equivalent, as shown by the low number of children with BPVC and headache in our sample (17.5%) [3,6,36,37,38,39].

BPPV is considered frequent in adults and can affect up to one-third of adults—in half of cases of vertigo [24,40]. In childhood it is not as representative, and many authors agree that it is rare outside of a traumatic context, although it is well described in children over 5 years old [2,37,41]. Children with BPPV made up 10.2% of the total cases of vertigo, with a mean age of 11 ± 3 years: the female to male ratio was 2:1. On the other hand, in our sample children with post-traumatic vertigo were only 1.5% of the total, although in the literature patients with post-concussion syndrome and dizziness reach 29.4% [42]. Similar to adults, BPPV appears during awakening in the morning or at night, which can cause a child to be convinced of having had nightmares. Symptoms and clinical characteristics of the BPPV are the same as those for adults, even if the feeling of vertigo is often better tolerated in children. Orthostatic hypotension is present during the pre-puberty or puberty period in females and is due to insufficient adaptation of the cardiovascular system to sudden changes of position of the body, mostly in case of poor physical activities. Orthostatic hypotension can cause quick orthostatic vertigo episodes (seconds to minutes) in 3 to 9% of symptomatic children, with possible consequent headaches and weaknesses [43]; in some cases, it may follow imminent loss of consciousness and falling to the ground [14]. In our sample, it was present in 4% of cases among who 16 of 19 patients were females.

Ménière’s disease is described as a very rare condition in a pediatric age (1.1% in literature; 0.8% in our sample) compared to adults (13.8%) and always occurs after 8 years of age [4,40,44]. However, many authors believe that the incidence of Ménière’s disease is underestimated in children [26,45]. In early phases of the disease, it is difficult to differentiate it from vestibular migraine [46,47].

The prevalence of psychogenic vertigo varies from 2.5% and 14% and is more frequent in girls between 8 and 10 years [14,48,49]. It has been found in 8.5% of the cases in our sample as the fifth most frequent cause of vertigo, with no difference in gender and age with the other forms of vertigo. The association between vertigo, and cognitive and psychiatric conditions is well known, increases with age, and is probably underestimated due to the lack of accurate research in all emotional aspects of vertigo in childhood [50].

Furthermore, 4.4% of the children in our sample had balance disorders caused by occlusal or temporomandibular dysfunction. An abnormal descending muscular interference could be considered as the cause of secondary proprioceptive postural dysfunction, with complete symptom recovery after dental treatment.

Vestibular vertigo is never associated with loss of consciousness, unlike epilepsy vertigo. In fact, seizure can be anticipated by aura, with suggestive signs in 2.8% of cases: visual or auditory hallucinations, neurological signs, imbalance, loss of consciousness, abnormal eye movements, no neurovegetative signs, no vertigo [27]. In our sample, epileptic vertigo accounted for 2.5% of cases.

Table 2, Table 3 and Table 4 report the clinical history, clinical examination results, and instrumental vestibular testing in our sample, which are key in improving diagnostic accuracy. From our analysis some considerations can be drawn. Family history for migraine is particularly significant in vestibular migraine and BPVC (respectively 57% and 35% in our sample); moreover, previous headache episodes are often reported in vestibular migraine, ophthalmologic disorders, and epileptic and post-traumatic vertigo (respectively 77%, 30.4%, 33.3%, and 42.9% in our sample); motion sickness can be associated with vestibular migraine and postural disorders (respectively 19% and 18.2% of our sample); neurovegetative symptoms are frequent in vestibular loss, migraine vertigo, ophthalmological disorders, and Meniere’s disease (respectively 42.4%, 30%, 26.1%, and 25% in our sample); hearing disorders are reported in otitis media and Meniere’s disease (respectively 20% and 75% in our sample) (Table 2). Nystagmus detection was uncommon in all dizzy patients evaluated, while vestibulospinal signs were frequent (Table 3). Our data suggest the importance of instrumental vestibular testing in children to increase accuracy of nystagmus detection. Moreover, vestibulospinal signs must be recognized in all dizzy patients. RCT and VHIT provide accurate functional vestibular information only in case of vestibular loss and Meniere’s disease and not in other conditions (Table 4).

The data collection in Table 2, Table 3 and Table 4 suggests that the accuracy of diagnosis of each form of vertigo is increased by performing detailed collection of clinical history, accurate clinical examination (both nystagmus and vestibulospinal signs) and adequate instrumental testing (at least RCT, better VHIT). For this reason, any neurotologic evaluation in children must be considered even if it is time-consuming, because a correct diagnostic process can prevent more complex and expensive investigations which may still be necessary in a few selected cases.

In fact, potentially life-threatening disorders that occur with vertigo are not very frequent; in these cases, vertigo is never an isolated feature, but is part of a much more complex clinical corollary [30]. Pathologies called into question are cranial trauma with temporal fracture, cardiovascular disease such as arrhythmias (i.e., prolonged QTc) and arterial hypertension, CNS pathologies such as posterior cranial fossa tumors (less than 1% of the cases), demyelinating diseases, and seizures [10,27]. Unlike other author’s reports, we did not find posterior cranial fossa tumors, although we did observe three patients affected by brain hemorrhage, multiple sclerosis, and L-Dopa responsive dystonia. In these cases, complete neurological evaluation, brain–brainstem MRI/CT imaging and EEG are indicated to differentiate posterior cranial fossa lesions, demyelinating diseases, and seizures [29,51].

Audiological evaluation must be considered in all dizzy children, since otitis media with effusion and internal ear malformations are possible conditions that cause imbalance in about 2% of cases (2.1% in the literature; 3.2% in our sample) [36]. For this reason, audiological evaluation (otoacoustic emissions, tympanometry, pure-tone audiometry, and ABR in selected cases) must always be assessed in all children affected by vestibular function impairment.

## 5. Conclusions

The prevalence of balance disorders in pediatric subjects is widely reported in the literature with a prevalence of about 10% in most clinical studies. A psychological, affective, physical, and relational impairment of a child can be a consequence of an incorrect or incomplete diagnostic and/or therapeutic work up.

Unlike in the literature where vestibular migraine and BPVC are considered the most frequent causes of vertigo in a pediatric age and undefined causes often exceed 10%, in our sample vestibular loss was the most frequent cause observed, and the cause of vertigo remained undefined in only 1.1% of cases. Diagnostic processes including detailed personal and family medical history, and accurate clinical and instrumental testing is the mainstay to identify the cause of vertigo. The improvement of knowledge in the vestibular field will allow for the introduction of new quick and non-invasive clinical and instrumental tests, effective for most of the patients at any age, while reserving more specific and/or invasive investigations to rare cases.

## Figures and Tables

**Table 1 children-08-01056-t001:** Vertigo in children and adolescents: distribution of causes in the entire group (*n* = 472) and by gender.

	Total			Females			Males		
Disease	No. of Children	Frequency (%)	Mean Age (years)	No. of Children	Frequency (%)	Mean Age (years)	No. of Children	Frequency (%)	Mean Age (years)
Vestibular loss	123	26.1%	12 ± 2	67	14.2%	12 ± 3	56	11.9%	12 ± 2
Vestibular migraine	100	21.2%	11 ± 3	59	12.5%	11 ± 3	41	8.7%	12 ± 2
BPPV	48	10.2%	11 ± 3	32	6.8%	11 ± 3	16	3.4%	10 ± 3
BPVC	40	8.5%	5 ± 3	27	5.7%	6 ± 3	13	2.8%	4 ± 2
Psychogenic vertigo	40	8.5%	12 ± 3	23	4.9%	12 ± 3	17	3.6%	12 ± 3
Ophthalmological disorders	23	4.9%	11 ± 3	12	2.5%	11 ± 3	11	2.3%	11 ± 3
Dental disorders	21	4.4%	10 ± 2	11	2.3%	11 ± 2	10	2.1%	10 ± 2
Hemodynamic orthostatic	19	4.0%	12 ± 3	16	3.4%	13 ± 2	3	0.6%	11 ± 3
Otitis media	15	3.2%	9 ± 4	7	1.5%	8 ± 4	8	1.7%	9 ± 4
Epileptic vertigo	12	2.5%	12 ± 2	8	1.7%	13 ± 2	4	0.8%	12 ± 2
Postural disorders	12	2.3%	11 ± 3	7	1.5%	12 ± 2	5	1.1%	10 ± 3
Post traumatic vertigo	7	1.5%	11 ± 3	3	0.6%	11 ± 3	4	0.8%	12 ± 3
Ménière’s disease	4	0.8%	9 ± 4	3	0.6%	9 ± 4	1	0.2%	6
Other neurological diseases	3	0.6%	8 ± 5	2	0.4%	10 ± 6	1	0.2%	6
No diagnosis	5	1.1%	11 ± 5	4	0.8%	12 ± 5	1	0.2%	8
Total	472		11 ± 3	281	59.5%	11 ± 3	191	40.5%	11 ± 3

BPPV: benign paroxysmal positional vertigo. BPVC: benign paroxysmal vertigo of children.

**Table 2 children-08-01056-t002:** Family history of migraine and distribution of symptoms according to cause of vertigo in children (No. of children with positive results).

Disease	No. of Children Affected	Family History for Migraine	Headache Episodes	Motion Sickness	Neurovegetative Symptoms	Hearing Disorders
Vestibular loss	123	13.6% (17)	27.2% (34)	8.8% (11)	42.4% (53)	6.4% (8)
Vestibular migraine	100	57% (57)	77% (77)	19% (19)	30% (30)	7% (7)
BPPV	48	22.9% (11)	14.6% (7)	6.3% (3)	22.9% (11)	4.2% (2)
BPVC	40	35% (14)	17.5% (7)	7.5% (3)	17.5% (7)	- (0)
Psychogenic vertigo	40	7.5% (3)	22.5% (9)	10% (4)	20% (8)	7.5% (3)
Ophthalmological disorders	23	8.7% (2)	30.4% (7)	8.7% (2)	26.1% (6)	4.3% (1)
Dental disorders	21	14.3% (3)	23.8% (5)	4.8% (1)	9.5% (2)	- (0)
Hemodynamic orthostatic	19	5.3% (1)	21.1% (4)	5.3% (1)	15.8% (3)	5.3% (1)
Otitis media	15	6.7% (1)	20% (3)	- (0)	6.7% (1)	20% (3)
Epileptic vertigo	12	- (0)	33.3% (4)	- (0)	- (0)	8.3% (1)
Postural disorders	12	- (0)	27.3% (3)	18.2% (2)	18.2% (2)	9.1% (1)
Post traumatic vertigo	7	42.9% (3)	42.9% (3)	14.3% (1)	- (0)	14.3% (1)
Ménière’s disease	4	25% (1)	25% (1)	- (0)	25% (1)	75% (3)
Other neurological diseases	3	- (0)	- (0)	- (0)	- (0)	- (0)
No diagnosis	5	20% (1)	20% (1)	- (0)	20% (1)	40% (2)

BPPV: benign paroxysmal positional vertigo. BPVC: benign paroxysmal vertigo of children.

**Table 3 children-08-01056-t003:** Percentage of signs revealed by clinical examination according to cause of vertigo in children (No. of children with positive results is reported in parenthesis).

Disease	No. of Children Affected	Spontaneous Nystagmus	Positional Nystagmus	Positioning Paroxysmal Nystagmus	Vestibulospinal Signs
Vestibular loss	123	20.8% (26)	13.6% (17)	0.8% (1)	29.6% (37)
Vestibular migraine	100	- (0)	4% (4)	- (0)	5% (5)
BPPV	48	- (0)	4.2% (2)	47.9% (23)	8.3% (4)
BPVC	40	- (0)	- (0)	- (0)	- (0)
Psychogenic vertigo	40	- (0)	- (0)	- (0)	7.5% (3)
Ophthalmological disorders	23	- (0) *	4.8% (1)	- (0)	9.5% (2)
Dental disorders	21	- (0)	4.8% (1)	- (0)	9.5% (2)
Hemodynamic orthostatic	19	- (0)	- (0)	- (0)	- (0)
Otitis media	15	- (0)	- (0)	- (0)	6.7% (1)
Epileptic vertigo	12	- (0)	8.3% (1)	- (0)	25% (3)
Postural disorders	12	- (0)	- (0)	- (0)	27.3% (3)
Post traumatic vertigo	7	- (0)	- (0)	- (0)	- (0)
Ménière’s disease	4	50% (2)	- (0)	- (0)	75% (3)
Other neurological diseases	3	- (0)	- (0)	- (0)	66.6% (2)
No diagnosis	5	- (0)	- (0)	- (0)	- (0)

BPPV: benign paroxysmal positional vertigo. BPVC: benign paroxysmal vertigo of children. * Two children were affected by congenital nystagmus.

**Table 4 children-08-01056-t004:** Percentage of abnormal results of RCT and VHIT according to cause of vertigo in children (No. of positive test/No. of children tested is reported in parenthesis).

Disease	No. of Children Affected	Abnormal Results of RCT	Abnormal Results of VHIT
Vestibular loss	123	89.7% (35/39)	71.6% (58/81)
Vestibular migraine	100	- (0/15)	- (0/52)
BPPV	48	- (0/3)	- (0/8)
BPVC	40	- (0/2)	- (0/9)
Psychogenic vertigo	40	- (0/3)	- (0/19)
Ophthalmological disorders	23	- (0/1)	- (0/12)
Dental disorders	21	- (0/1)	- (0/14)
Hemodynamic orthostatic	19	- (0/2)	- (0/5)
Otitis media	15	- (0/1)	- (0/7)
Epileptic vertigo	12	- (0/1)	- (0/7)
Postural disorders	12	-	- (0/6)
Post traumatic vertigo	7	- (0/1)	- (0/5)
Ménière’s disease	4	100% (2/2)	- (0/1)
Other neurological diseases	3	-	- (0/1)
No diagnosis	5	-	- (0/5)

RCT: rotatory chair test. VHIT: video hit impulse test. BPPV: benign paroxysmal positional vertigo. BPVC: benign paroxysmal vertigo of children.

## Data Availability

The data presented in this study are available on request from the corresponding author. The data are not publicly available due to privacy.

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
