# Peer review of "Prevalence of Pediatric and Adolescent Balance Disorders: Analysis of a Mono-Institutional Series of 472 Patients"

_children, 2021, doi:10.3390/children8111056_

Round 1
Reviewer 1 Report
The paper shows interesting data about the distribution of different vestibular disorders in a large cohort of children assessed for vertigo complaints.
These are my concerns about the manuscript:
-The authors mainly showed the percentages of the various pathologies found and the results of the vestibular examinations/testing performed at their Institution. For this reason, I would modify the title emphasizing the aspect of the prevalence of vestibular disorders rather than the diagnostic process (I do not think the authors are really providing a diagnostic work-up to the reader).
-Did you visit patients with hearing impairment (of course apart from those affected by Meniere’s disease)? Did you find any bilateral areflexia and/or cases of vestibular loss related to hearing impairment (e.g. malformations, congenital deafness….)?
- All your cases of vestibular loss seemed to be related only to vestibular neuritis and only the 20.8% of patients with vestibular loss showed a spontaneous nystagmus (so this is the percentage of patients seen in the acute phase of vestibular neuritis). You stated that “Abnormal results of VHIT, including reduced HIMP and SHIMP VOR amplitude and/or presence of refixation overt/covert saccades, were observed in 71.6% of vestibular loss.” Could you explain this low percentage of abnormal VHIT that you found? Was the remaining percentage represented by patients with an abnormal RCT? If not, how was the diagnosis of vestibular loss made? Based on the clinical history?
-You reported “Moreover, it is well known that children with vertigo and normal vestibular function could present various vergence dysfunctions [32, 33] and that 10% of the children aged 5-6 years suffer from dizziness due to eye problems (4.9% in our sample)”. However, your cohort presented a mean age of 11±3 years. Were all your patients affected by ophthalmological disorders in the same age range of the ones reported by published data?
Author Response
Dear Reviewer,
Thank you for your precious comments and suggestions, which will surely add scientific value to our work. Here below you find our point-by-point replies.
- I would modify the title emphasizing the aspect of the prevalence of vestibular disorders rather than the diagnostic process).
We agree with your comment, and we had modified the title
- Did you visit patients with hearing impairment (of course apart from those affected by Meniere’s disease)? Did you find any bilateral areflexia and/or cases of vestibular loss related to hearing impairment (e.g. malformations, congenital deafness….)?
Children sent only for hearing impairment were excluded from the analysis. This is now better specified in criteria of exclusion
- All your cases of vestibular loss seemed to be related only to vestibular neuritis and only the 20.8% of patients with vestibular loss showed a spontaneous nystagmus (so this is the percentage of patients seen in the acute phase of vestibular neuritis). You stated that “Abnormal results of VHIT, including reduced HIMP and SHIMP VOR amplitude and/or presence of refixation overt/covert saccades, were observed in 6%of vestibular loss.” Could you explain this low percentage of abnormal VHIT that you found? Was the remaining percentage represented by patients with an abnormal RCT? If not, how was the diagnosis of vestibular loss made? Based on the clinical history?
Based on your observation, we carefully revisited our entire database, and we found a typo. In fact, the RCT was performed in 39 cases (not 9), of which 35 were positive (not 5). In 4 cases the Vestibular Loss diagnosis was made without any spontaneous nystagmus/vHIT/RCT alteration. Reconsidering their case-by-case data, we found that the diagnosis was made exclusively based on symptoms and altered HST/Unterberger test. We apologize for the big mistake, and we thank you for your pertinent note.
- You reported “Moreover, it is well known that children with vertigo and normal vestibular function could present various vergence dysfunctions [32, 33] and that 10% of the children aged 5-6 years suffer from dizziness due to eye problems (4.9% in our sample)”. However, your cohort presented a mean age of 11±3 years. Were all your patients affected by ophthalmological disorders in the same age range of the ones reported by published data?
We are sorry that the sentence is misleading. In the revised manuscript we corrected the sentence.
Reviewer 2 Report
The article represents an interesting mix between the personal experience of a group of researchers who are experts in the field and a critical reading of the literature. Interest in pediatric vestibology is on the rise and this article can certainly make an important contribution to those approaching the subject.
I believe the article deserves prompt publication, but actually needs minor revisions.
I think some more extensive considerations on neurological assessment and exclusion criteria are needed.
Proper formatting of the tables is absolutely necessary. The tables are illegible, the title of Table 1 is missing, it would be useful to insert a short caption for each table.
Line 90: BPVC should be named before using abbreviation, which happens in line 103
Author Response
Dear Reviewer,
thank you for your evaluable comments, which will add scientific value to our work.
Here below you find our point-by-point replies.
- I think some more extensive considerations on neurological assessment and exclusion criteria are needed.
We added specifications on neurological assessment of our institution in case of negative vestibular assessment and explained exclusion criteria in a more exhaustive way
- Proper formatting of the tables is absolutely necessary. The tables are illegible, the title of Table 1 is missing, it would be useful to insert a short caption for each table.
We are sorry for the formatting of the tables due to a conversion of the Word file to pdf. We took advantage of your comment to explain the table in a better way
- Line 90: BPVC should be named before using abbreviation, which happens in line 103
Thank you for your comment, and we modified the text accordingly